# Prevalence and predictors of suboptimal glycemic control among patients with type 2 diabetes mellitus in northern Thailand: A hospital-based cross-sectional control study

**Fartima Yeemard[1], Peeradone Srichan[1,2], Tawatchai Apidechkul[1,2]\*, Naphat Luerueang[3], Ratipark Tamornpark[1,2], Suphaphorn Utsaha[2]**

**1** Center of Excellence for the Hill tribe Health Research, Mae Fah Luang University, Chiang Rai, Thailand, **2** School of Health Science, Mae Fah Luang University, Chiang Rai, Thailand, **3** Mae Lao District Public Health Office, Chiang Rai, Thailand

\* Tawatchai.api@mfu.ac.th

## Abstract

### Background

Suboptimal glycemic control among patients with type 2 diabetes mellitus (DM) is a significant public health problem, particularly among people living with poor education and economic statuses, including those with a unique dietary culture. This study aimed to estimate the prevalence and identify the factors associated with suboptimal glycemic control among patients with type 2 DM during the coronavirus disease-2019 (COVID-19) pandemic.

### Methods

A hospital-based cross-sectional study was used to elicit information from DM patients attending six hospitals located in Chiang Rai Province, northern Thailand, between February and May 2021. A validated questionnaire and 5 mL blood specimens were used as the research tools. Glycated hemoglobin (HbA1c) greater than 7.0% among DM patients at least two years after diagnosis was defined as suboptimal glycemic control. Chi-square tests and logistic regression were used to identify the associations between variables at the significance level α = 0.05.

### Results

A total of 967 patients were recruited for this study; 54.8% 530 had suboptimal glycemic control, 58.8% were female, 66.5% were aged 50-69 years, and 78.5% were married (78.5%). Six variables were found to be associated with suboptimal glycemic control in multivariable logistic regression. Participants aged <49, 50-59, and 60-69 years had 3.32 times (95% CI = 1.99-5.53), 2.61 times (95% CI = 1.67-4.08), and 1.93 times (95% CI = 1.26-2.95) greater odds of having suboptimal glycemic control, respectively, than those aged ≥70 years. Married individuals had 1.64 times (95% CI = 1.11-2.41) greater odds of having suboptimal glycemic control than those ever married. Participants who consumed sticky rice

**Data Availability Statement:** All relevant data are within the paper and its Supporting Information files.

**Funding:** This research was supported by the Center of Excellence for the Hill tribe Health Research, Mae Fah Lung University, Thailand (Grant Number 1-2021). The funders had no role in study design, data collection and analysis, decision to publish, or preparation of the manuscript.

**Competing interests:** The authors have declared that no competing interests exist.

had 1.61 times (95% CI = 1.19-2.61) greater odds of having suboptimal glycemic control than those who did not consume sticky rice in daily life. Participants who had been diagnosed with DM for 11-20 years and ≥21 years had 1.98 times (95% CI = 1.37-2.86) and 2.46 times (1.50-4.04) greater odds of having suboptimal glycemic control, respectively, than those who had been diagnosed ≤ 10 years. Participants who had experienced forgetting to take their medication had 2.10 times (95% CI = 1.43-3.09) greater odds of having suboptimal glycemic control than those who did not, and those who had their medical expenses covered by the national scheme had 2.67 times (95% CI = 1.00-7.08) greater odds of suboptimal glycemic control than those who self-paid.

## Conclusion

Effective health interventions to control blood glucose among DM patients during ongoing treatment are urgently required. The interventions should focus on patients aged less than 69 years, marital status, forgetting to take their medication, and a longer time since diagnosis, including reducing their sticky rice consumption. The effects of copayments should also be considered.

## Introduction

Diabetes mellitus (DM) with suboptimal glycemic control is one of the major health problems among people around the world, especially those who are living in low- and middle-income countries, including Thailand [1]. The World Health Organization (WHO) estimated that 422 million patients globally are affected by DM [2] and 1.5 million deaths were directly caused by DM, and an additional 3 million deaths were secondarily caused by DM in 2019, primarily from suboptimal glycemic control [1]. In addition to the lives lost from the disease, it also adversely impacts economics, especially due to the resources required for the treatment and care of DM patients for many years [3, 4]. Moreover, the quality of life of those who have this condition and their family members are also reduced due to the need for constant care during everyday life [5, 6]. A significant proportion of DM patients are properly diagnosed and well cared for, but the effectiveness of maintaining or controlling their blood glucose is very poor, particularly in Thailand [7].

Thailand is defined as an upper-middle-income country with approximately 67 million people [8]. The WHO reported that the prevalence of DM among the Thai population aged 30 years and over was 9.6% (9.1% for men and 10.1% for women [9]. Thailand has a policy and strategic plan for reducing diabetes problems, but it has no programs to address overweight and obesity, including physical inactivity problems [9]. The majority of Thai people work in the agricultural sector, including people in northern Thailand [10]. People living in northern Thailand have some special cultures and lifestyles, including unique cooking practices. Sticky rice is the main dish in daily life, and consuming oily noodles for lunch is also common [11]. Since 2018, the northern region has been classified as having the greatest proportion of people aged ≤60 years in the country, namely an aging society [12], while its economic level has been ranked at a lower level than the central and southern regions [13].

In 2020, the Ministry of Public Health, Thailand estimated that 5 million people have developed DM among the Thai population or one in every eleven among people aged 15 years and over [14]. People living in the northern region of Thailand, where the local people have special

cultures and lifestyles, including the consumption of sticky rice and popular oily dishes in their daily lives [15], have been reported to have one of the highest DM prevalences in Thailand [16]. Moreover, 200 deaths every day have been attributed to DM [14]. Aekplakorn et al. [7] reported that the prevalence of DM among the Thai population was 10.8% among individuals aged 20 years and over, and the overall suboptimal glycemic control was 29.5% (23.9% among men and 35.7% among women). DM patients with poor economic and educational statuses have a higher risk of having suboptimal glycemic control. In addition, a large amount of money has been allocated for DM case management in Thailand [17], particularly to those who have a problem of suboptimal glycemic control while living with a unique culture. A few scientific studies are available to understand the prevalence and factors contributing to suboptimal glycemic control among DM patients in northern Thailand, especially during the coronavirus disease-2019 (COVID-19) crisis, when the schedules of health care services were modified to fit the situation in hospitals.

The aims of this study were to estimate the prevalence of suboptimal glycemic control and identify factors associated with suboptimal glycemic control among Thai people living in northern Thailand.

## Materials and methods

### Study design and study setting

A cross-sectional study was used to collect data from participants who were DM patients attending six hospitals: Mae Lao Hospital, Mae Chan Hospital, Wiang Chiang Rung Hospital, Phan Hospital, Wiang Chai Health Promoting Hospital, and Mae Lao Health Promoting Hospital, which were selected by a random method from among 18 hospitals in Chiang Rai Province, Thailand.

### Study population and eligible population

DM patients who attended DM clinics in the six hospitals were the study population. Those who attended a clinic between February and May 2021 and had been diagnosed at least two years prior with DM and treated met the inclusion criteria. However, those who had severe illness, were admitted to the inpatient department, pregnant, or could not provide essential information were excluded from the study.

### Sample size

The sample size was calculated based on the standard formula for a cross-sectional design [18]; $n = [Z^2_{\alpha/2}P\,(1\text{-}P)]/d^2$, where the Z = value from the standard normal distribution corresponded to the desired confidence level (Z = 1.96 for 95% CI), P = the expected true proportion (P = 0.30 [7], and d = precision (d = 0.03); after adding 5% to account for any error in the study, 941 participants were needed for the analysis.

### Research instruments

A questionnaire developed by the researcher was used for data collection. It consisted of six parts. In part one, seven items were used to collect physical examination and laboratory data, such as weight, height, blood pressure, triglyceride level, low-density lipoprotein cholesterol (LDL-C), and high-density lipoprotein cholesterol (HDL-C). In part two, twenty questions were used to collect general information, such as sex, age, religion, tribe, and education. In part three, eight questions were used to collect data regarding health behaviors such as smoking, alcohol use, having tea, having coffee, etc. In part four, five questions about the stress test

(ST-5) [19] were used to detect the level of stress. In part five, ten questions were used to detect knowledge about DM prevention and control. In the last part, ten questions were used to detect attitudes toward DM prevention and control (S1 Questionnaire).

A five mL blood specimen was drawn to detect HbA1c levels and other lipid profiles, such as LDL-C, HDL-C, and triglycerides.

## Validated questionnaire

The validity and reliability of the questionnaire were detected by different methods. Item-objective congruence was used to detect the validity of the questionnaire. Using this method, three external experts assessed the congruence between the questions and the context of the study, including the objectives of the study. The experts provided the score for each item: "-1" means that the question is not related to the content and objectives of the study; "0" means that the question is related to the content and objectives of the study but requires improvement before use; and "+1" means that the question is relevant to the content and the objectives of the study and does not require any improvement. The scores from the experts were pooled and divided by three before interpretation. If the questions had an average score less than 0.5, the questions were deleted from the questionnaire. If the questions were scored between 0.5-0.7, they were improved before being added to the questionnaire. The items with a score greater than 0.7 were included in the final questionnaire.

The questionnaire was piloted with 20 people who had similar characteristics to the study population and attended the DM clinic at Mae Sai Hospital. In this step, the feasibility, proper words or sentences used, and ordering of the questions, including the reliability, were analyzed. Finally, Cronbach's alpha was found to be 0.72 for the final questionnaire.

## Measures

Body weight index (BMI) is a standard used by the WHO that is classified into three categories: underweight ($\leq$18.49), normal weight (18.50-24.99), and overweight ($\geq$25.00) [20]. Stress was classified into three categories [21]: low ($\leq$4 scores), moderate (5-7 scores), and high ($\geq$8 scores). The triglycerides were classified into two major groups according to guidelines from the World Health Organization [22]: normal (<150 mg/dL) and high ($\geq$150 mg/dL). According to the World Health Organization (WHO), low-density lipoprotein cholesterol (LDL-C) levels were classified into two main groups: normal (<100 mg/dL) and high ($\geq$100 mg/dL) [22]. High-density lipoprotein cholesterol (HDL-C) was classified into two groups: low (<40 mg/dL) and normal ($\geq$40 mg/dL) [22]. A glycated hemoglobin (HbA1c) level $\geq$ 7 was defined as suboptimal glycemic control [23].

## Data gathering procedures

The hospital directors and the chiefs of DM clinics were contacted and we explained the study objectives, including the procedures of data collection, after obtaining ethical approval for conducting this research from the Chiang Rai Provincial Public Health Office. On the day of data collection, all DM patients who attended the clinic were invited to join the study on a voluntary basis. Those who were willing to participate in the study were given an explanation of the study objectives and the procedure of data and blood specimen collection. After signing the written consent form, the participants were asked to fill out the questionnaire. For those who could not use Thai, the participants were requested to use fingerprints to sign the consent form. Researchers completed the questionnaire for those who could not use Thai after obtaining the information from the participants. A 5 mL blood sample was collected by medical technicians who had validated licensing. The participating DM patients were asked to fast (nothing

per oral (NPO)) for 12 hrs before blood collection. All specimens were stored properly and transferred to the Mae Fah Laung Medical Laboratory Center for analysis on the same day.

## Statistical analysis

The data were coded in an Excel sheet followed by fact-checking before transfer into SPSS with the SPS program (version 24, Chicago, IL). Descriptive statistics were used to describe the general characteristics of the participants. Percentages are used to describe the categorical data, while other continuous data are described by the mean and standard deviation (SD) for a normal distribution, and the median and interquartile range (IQR) for the skewed data. Chi-square and Fisher exact tests were used to detect the associations. Logistic regression was used to detect the associations between the independent variables and dependent variable at a significance of $\alpha = 0.05$. Selection variable into the model, the "Enter" mode was used. The Cox-Snell $R^2$ and Nagelkerke $R^2$ and the Hosmer-Lemshow were used to determine the fit of the model in all steps. These variables found to be significant in the univariable logistic model were considered to be put into the multivariable model. Before fitting the final multivariable logistic model, some variables were controlled as the confounder factors before interpretation.

## Ethics approval and consent to participate

Consent to participate, all study instruments and procedures were approved by the Ethics Committee for Human Research, Chiang Rai Provincial Public Health Office, Chiang Rai, Thailand (CRPHO No.6/2465). All participants received an oral and written explanation and provided their consent before a voluntary agreement was witnessed and documented by signature or fingerprint.

## Results

### General characteristics of the participants

A total of 967 cases (530 suboptimal glycemic control (54.8%) and 437 controlled blood glucose (45.2%)) were recruited from 6 hospitals: 294 cases (30.4%) from Phan Hospital, 301 cases (31.1%) from Mae Chan Hospital, 208 cases (21.5%) from Mae Lao Hospital, 84 cases (8.7%) from Wiang Chiang Rung Hospital, 40 cases (4.1%) from Mae Lao Health Promoting Hospital, and 40 cases (4.1%) from Wiang Chai Health Promoting Hospital.

More than half were women (58.8%), 66.5% were aged 50-69 years (mean = 58.7, min = 18, max = 97, and SD = 11.3), and 99.1% held Thai identification cards (IDs). The majority were married (78.5%), Buddhist (99.5%), graduated from primary school (67.1%), worked as farmers (36.4%), and had an annual family income less than 50,000 baht (68.7%), with a median of 40,000 baht and an IQR of 30,000 baht (Table 1).

Four factors were significantly different between the uncontrolled and controlled blood glucose groups: sex (p-value = 0.017), age (p-value<0.001), marital status (p-value = 0.018), and family debt (p-value = 0.002) (Table 1).

### Health behaviors, knowledge and attitudes toward DM prevention and control

Almost one-fifth smoked (16.2%), 15.9% used alcohol, and more than half did not exercise in their daily life. A large proportion cooked their own food (88.4%), 31.9% regularly ate sticky rice, 15.2% drank tea, and 31.7% drank coffee. One-fourth (25.6%) had stress between moderate-to-high levels, 47.8% had knowledge regarding DM prevention and control at low-to-

**Table 1. General characteristics of participants.**

| Characteristics | Total | | Suboptimal glycemic control | | | | $\chi^2$ | p-value |
|---|---|---|---|---|---|---|---|---|
| | | | Yes | | NO | | | |
| | n | % | n | % | n | % | | |
| **Total** | **967** | **100.0** | **530** | **54.8** | **437** | **45.2** | N/A | A/A |
| **Sex** | | | | | | | | |
| Male | 398 | 41.2 | 200 | 50.3 | 198 | 49.7 | 5.67 | 0.017* |
| Female | 569 | 58.8 | 330 | 58.0 | 239 | 42.0 | | |
| **Age** (years) | | | | | | | | |
| ≤49 | 187 | 19.3 | 66 | 35.3 | 121 | 64.7 | 27.49 | <0.001* |
| 50-59 | 274 | 28.3 | 163 | 59.5 | 111 | 40.5 | | |
| 60-69 | 369 | 38.2 | 195 | 52.8 | 174 | 47.2 | | |
| ≥70 | 137 | 14.2 | 51 | 37.2 | 86 | 62.8 | | |
| **Tribe** | | | | | | | | |
| Hill tribe | 28 | 2.9 | 18 | 64.3 | 10 | 35.7 | 1.04 | 0.307 |
| Thai people | 939 | 97.1 | 512 | 54.5 | 427 | 45.5 | | |
| **Thai ID card** | | | | | | | | |
| No | 9 | 0.9 | 7 | 77.8 | 2 | 22.2 | 1.93 | 0.164 |
| Yes | 958 | 99.1 | 523 | 54.6 | 435 | 45.4 | | |
| **Marital status** | | | | | | | | |
| Single | 70 | 7.2 | 43 | 61.4 | 27 | 38.6 | 8.03 | 0.018* |
| Married | 759 | 78.5 | 426 | 56.1 | 333 | 43.9 | | |
| Ever married | 138 | 14.3 | 61 | 44.2 | 77 | 55.8 | | |
| **Religion** | | | | | | | | |
| Buddhist | 962 | 99.5 | 528 | 54.9 | 434 | 45.1 | 0.44 | 0.505[a] |
| Christian or Muslim | 5 | 0.5 | 2 | 40.0 | 3 | 60.0 | | |
| **Education** | | | | | | | | |
| No education | 122 | 12.6 | 58 | 47.5 | 64 | 52.5 | 4.11 | 0.128 |
| Primary school | 649 | 67.1 | 356 | 54.9 | 293 | 45.1 | | |
| Secondary school and higher | 196 | 20.3 | 116 | 59.2 | 80 | 40.8 | | |
| **Occupation** | | | | | | | | |
| Unemployed | 283 | 29.3 | 147 | 51.9 | 136 | 48.1 | 3.41 | 0.331 |
| Famer | 352 | 36.4 | 188 | 53.4 | 164 | 46.6 | | |
| Employed | 224 | 23.2 | 130 | 58.0 | 94 | 42.0 | | |
| Trade and government officer | 108 | 11.2 | 65 | 60.2 | 43 | 39.8 | | |
| **Annual income** (baht) | | | | | | | | |
| ≤ 50,000 | 664 | 68.7 | 366 | 55.1 | 298 | 44.9 | 1.67 | 0.433 |
| 50,001-100,000 | 189 | 19.5 | 97 | 51.3 | 92 | 48.7 | | |
| ≥100,001 | 114 | 11.8 | 67 | 58.8 | 47 | 41.2 | | |
| **Family debt** | | | | | | | | |
| No | 566 | 58.5 | 334 | 59.0 | 232 | 41.0 | 9.72 | 0.002* |
| Yes | 401 | 41.5 | 196 | 48.9 | 205 | 51.1 | | |
| **Family members** (people) | | | | | | | | |
| ≤ 4 | 828 | 85.6 | 444 | 53.6 | 384 | 46.4 | 3.26 | 0.071 |
| ≥ 5 | 139 | 14.4 | 86 | 61.9 | 53 | 38.1 | | |
| **Living with** | | | | | | | | |
| Alone | 61 | 6.3 | 28 | 45.9 | 33 | 54.1 | 2.383 | 0.497 |
| Spouse | 589 | 60.9 | 330 | 56.0 | 259 | 44.0 | | |
| Child | 216 | 22.3 | 118 | 54.6 | 98 | 45.4 | | |
| Relatives | 101 | 10.4 | 54 | 53.5 | 47 | 46.5 | | |

*Significance level α = 0.05

[a] Fisher's exact test

moderate levels, and 86% had attitudes toward DM prevention and control at poor-to-moderate levels (Table 2).

Seven variables were found to be different health behaviors between the uncontrolled and controlled blood glucose groups: smoking (p-value<0.001, preparing food (p-value = 0.019), type of rice eaten daily (p-value<0.001), drinking tea (p-value<0.001), drinking coffee (p-value<0.001), and attitude toward DM prevention and control (p-value = 0.001) (Table 2).

**Table 2. Substance use and health behaviors among the participants.**

| Health behaviors | Total | | Suboptimal glycemic control | | | | $\chi^2$ | p-value |
|---|---|---|---|---|---|---|---|---|
| | | | Yes | | NO | | | |
| | n | % | n | % | n | % | | |
| **Smoking** | | | | | | | | |
| No | 810 | 83.8 | 423 | 52.2 | 387 | 47.8 | 13.47 | <0.001* |
| Yes | 157 | 16.2 | 107 | 68.2 | 50 | 31.8 | | |
| **Alcohol use** | | | | | | | | |
| No | 813 | 84.1 | 441 | 54.2 | 372 | 45.8 | 0.65 | 0.417 |
| Yes | 154 | 15.9 | 89 | 57.8 | 65 | 42.2 | | |
| **Exercise** | | | | | | | | |
| No | 499 | 51.6 | 198 | 39.7 | 301 | 60.3 | 95.42 | <0.001* |
| Sometime | 265 | 27.4 | 190 | 71.7 | 75 | 28.3 | | |
| Regular | 203 | 21.0 | 142 | 70.0 | 61 | 30.0 | | |
| **Preparing food** | | | | | | | | |
| Themselves | 855 | 88.4 | 457 | 53.5 | 398 | 46.5 | 5.49 | 0.019* |
| Buying | 112 | 11.6 | 73 | 65.2 | 39 | 34.8 | | |
| **Type of rice eaten daily** | | | | | | | | |
| Non-sticky rice | 659 | 68.1 | 332 | 50.4 | 327 | 49.6 | 16.38 | <0.001* |
| Sticky rice | 308 | 31.9 | 198 | 64.3 | 110 | 35.7 | | |
| **Frequency of having sticky rice per day** | | | | | | | | |
| One | 16 | 5.2 | 12 | 75.0 | 4 | 25.0 | 2.75 | 0.252 |
| Two | 22 | 7.1 | 17 | 77.3 | 5 | 22.7 | | |
| Three | 270 | 87.7 | 169 | 62.6 | 101 | 37.4 | | |
| **Drinking tea** | | | | | | | | |
| No | 820 | 84.8 | 425 | 51.8 | 395 | 48.2 | 19.33 | <0.001* |
| Yes | 147 | 15.2 | 105 | 71.4 | 42 | 28.6 | | |
| **Drinking coffee** | | | | | | | | |
| No | 660 | 68.3 | 323 | 48.9 | 337 | 51.1 | 28.91 | <0.001* |
| Yes | 307 | 31.7 | 207 | 67.4 | 100 | 32.6 | | |
| **Stress** (ST-5) | | | | | | | | |
| Low | 719 | 74.4 | 390 | 54.2 | 329 | 45.8 | 0.36 | 0.832 |
| Moderate | 159 | 16.4 | 90 | 56.6 | 69 | 43.4 | | |
| High | 89 | 9.2 | 50 | 56.2 | 39 | 43.8 | | |
| **Knowledge regarding DM prevention and control** | | | | | | | | |
| Low | 78 | 8.1 | 37 | 47.4 | 41 | 52.6 | 4.50 | 0.105 |
| Moderate | 384 | 39.7 | 201 | 52.3 | 183 | 47.7 | | |
| High | 505 | 52.2 | 292 | 57.8 | 213 | 42.2 | | |
| **Attitudes toward DM prevention and control** | | | | | | | | |
| Poor | 270 | 27.9 | 122 | 45.2 | 148 | 54.8 | 14.00 | 0.001* |
| Moderate | 562 | 58.1 | 329 | 58.5 | 233 | 41.5 | | |
| Positive | 135 | 14.0 | 79 | 58.5 | 56 | 41.5 | | |

*Significance level α = 0.05

## Experiences related to DM treatment and care and biomarkers

Among the participants, 26.7% were diagnosed with DM more than 10 years prior, 2.1% were self-paying for all medical expenses, 19.0% had experienced forgetting to take a medication, and 13.5% had experienced side effects while using a medication related to DM treatment. Sixty people had experienced wounds on their feet, 50.3% had diabetic nephropathy, and 53.5% had HT. A large proportion were overweight (65.3%), 44.8% had high LDL cholesterol levels, and 49.7% had high triglyceride levels (Table 3).

Nine (9) variables were found to be significantly different between the suboptimal glycemic control and the controlled blood glucose groups: the duration since the DM diagnosis (p-value = 0.001), experience of forgetting to take a medication (p-value<0.001), having side effects from DM medications (p-value<0.001), having wounds on the foot (p-value<0.001), diabetic nephropathy (p-value = 0.001), having HT (p-value = 0.001), BMI (p-value = 0.001), and LDL cholesterol (p-value = 0.007) (Table 3).

## Prevalence and factors associated with suboptimal glycemic control

The prevalence of suboptimal glycemic control was 54.8% (50.3% in men and 58.0% in women). The highest prevalence of suboptimal glycemic control was among those aged 50-59 (59.5%) and 60-69 years (58.0%).

In the model to identify socio-demographics that associated with suboptimal glycemic control among DM patients by univariable logistic regressions, five (5) variables were found to be associated with suboptimal glycemic control: sex, age, marital status, education, and family debt (Table 4).

Only two (2) variables were found to be associated with suboptimal glycemic control in multivariable logistic regression: age, and marital status. Participants aged <40, 40-49, 50-59, and 60-69 years had 3.32 times (95% CI = 1.99-5.53), 2.61 times (95% CI = 1.67-4.08), and 1.93 times (95% CI = 1.26-2.95) greater odds of having suboptimal glycemic control, respectively, than those aged ≥70 years. Married participants had 1.64 times (95% CI = 1.11-2.41) greater odds of having suboptimal glycemic control than those ever married (Table 4).

In the model to identify behavioral and psychological determinants associated with suboptimal glycemic control among DM patients by univariable logistic regression, six (6) variables were found to be associated with suboptimal glycemic control among the DM: smoking, exercise, preparing food in daily life, type of rice consumed daily, drinking tea, drinking coffee, and attitudes toward DM prevention and control (Table 5).

Only one variable was found to be associated with suboptimal glycemic control in multivariable logistic regression. Participants who consumed sticky rice in daily life had 1.61 times (95% CI = 1.19-2.61) greater odds of having suboptimal glycemic control than those who did not (Table 5).

In the model to identify medication and biochemical markers that associated with suboptimal glycemic control among DM patients by univariable logistic regressions, seven (7) variables were found to be associated with suboptimal glycemic control among the DM patients attending hospitals in northern Thailand: duration since the DM diagnosis, forgetting to take a medication, having side effects from taking medications, a history of having wounds on the feet, having diabetic nephropathy, and having elevated HT and LDL cholesterol (Table 6).

Only three (3) variables were found to be associated with suboptimal glycemic control in multivariable logistic regression: time since being diagnosed with DM, forgetting to take a medication, and medical expenses. Participants who had been diagnosed with DM 11-20 years ago and more than 20 years ago had 1.98 times (95% CI = 1.37-2.86) and 2.46 times (1.50-4.04) greater odds of having suboptimal glycemic control, respectively, than those who had been

**Table 3. DM experiences and biomarkers of the participants.**

| Health indicators | Total | | Suboptimal glycemic control | | | | $\chi^2$ | p-value |
|---|---|---|---|---|---|---|---|---|
| | | | Yes | | No | | | |
| | n | % | n | % | n | % | | |
| **Length of DM diagnosed** (year) | | | | | | | | |
| ≤ 10 | 709 | 73.3 | 362 | 51.1 | 347 | 48.9 | 15.16 | 0.001* |
| 11-20 | 172 | 17.8 | 113 | 65.7 | 59 | 34.3 | | |
| > 20 | 86 | 8.9 | 55 | 64.0 | 31 | 36.0 | | |
| **Medical expenses** | | | | | | | | |
| Covered by the national universal scheme | 947 | 97.9 | 519 | 54.8 | 428 | 45.2 | 0.00 | 0.986 |
| Self-paid | 20 | 2.1 | 11 | 55.0 | 9 | 45.0 | | |
| **Experience of forgetting to take a medication** | | | | | | | | |
| No | 783 | 81.0 | 398 | 50.8 | 385 | 49.2 | 26.29 | <0.001* |
| Yes | 184 | 19.0 | 132 | 71.7 | 52 | 28.3 | | |
| **Having side effects from DM medications** | | | | | | | | |
| No | 836 | 86.5 | 434 | 51.9 | 402 | 48.1 | 20.87 | <0.001* |
| Yes | 131 | 13.5 | 96 | 73.3 | 35 | 26.7 | | |
| **History of wounds on foot** | | | | | | | | |
| No | 907 | 93.8 | 482 | 53.1 | 425 | 46.9 | 16.39 | <0.001* |
| Yes | 60 | 6.2 | 48 | 80.0 | 12 | 20.0 | | |
| **Diabetic nephropathy** | | | | | | | | |
| No | 375 | 38.8 | 191 | 50.9 | 184 | 49.1 | 14.57 | 0.001* |
| Yes | 486 | 50.3 | 263 | 54.1 | 223 | 45.9 | | |
| Do not know | 106 | 11.0 | 76 | 71.7 | 30 | 28.3 | | |
| **Having HT** | | | | | | | | |
| No | 395 | 40.8 | 225 | 57.0 | 170 | 43.0 | 14.29 | 0.001* |
| Yes | 517 | 53.5 | 263 | 50.9 | 254 | 49.1 | | |
| Do not know | 55 | 5.7 | 42 | 76.4 | 13 | 23.6 | | |
| **BMI** (kg/m$^2$) | | | | | | | | |
| Underweight (≤ 18.50) | 53 | 5.5 | 27 | 50.9 | 26 | 49.1 | 14.03 | 0.001* |
| Normal weight (18.51-22.99) | 283 | 29.3 | 130 | 45.9 | 153 | 54.1 | | |
| Overweight (≥ 23.00) | 631 | 65.3 | 373 | 59.1 | 258 | 40.9 | | |
| **HDL cholesterol** (mg/dL) | | | | | | | | |
| Low (< 40) | 262 | 27.1 | 144 | 55.0 | 118 | 45.0 | 0.00 | 0.953 |
| Normal (≥ 40) | 705 | 72.9 | 386 | 54.8 | 319 | 45.2 | | |
| **LDL cholesterol** (mg/dL) | | | | | | | | |
| Normal (<100) | 534 | 55.2 | 272 | 50.9 | 262 | 49.1 | 7.22 | 0.007* |
| High (≥100) | 433 | 44.8 | 258 | 59.6 | 175 | 40.4 | | |
| **Triglycerides** (mg/dL) | | | | | | | | |
| Normal (< 150) | 486 | 50.3 | 253 | 52.1 | 233 | 47.9 | 2.98 | 0.084 |
| High (≥ 150) | 481 | 49.7 | 277 | 57.6 | 204 | 42.4 | | |

*Significance level α = 0.05

diagnosed ≤ 10 years prior. Participants who had experienced forgetting to take medication had 2.10 times (95% CI = 1.43-3.09) greater odds of having suboptimal glycemic control than those who did not, and those who were covered for their medical expenses by the national scheme had 2.67 times (95% CI = 1.00-7.08) greater odds of more suboptimal glycemic control than those who self-paid (Table 6).

**Table 4. Identifying socio-demographics that associated with suboptimal glycemic control among DM patients by univariable and multivariable logistic regressions.**

| Factors | Suboptimal glycemic control | | Univariable analysis | | | Multivariable analysis | | |
|---|---|---|---|---|---|---|---|---|
| | Yes (%) | No (%) | OR | 95% CI | p-value | AOR | 95% CI | p-value |
| Total | 530 (54.8) | 437 (45.2) | N/A | N/A | N/A | N/A | N/A | N/A |
| **Sex** | | | | | | | | |
| Male | 200 (50.3) | 198 (49.7) | 1.00 | | | | | |
| Female | 330 (58.0) | 239 (42.0) | 1.39 | 1.07-1.79 | 0.012* | | | |
| **Age** (years) | | | | | | | | |
| ≤49 | 66 (35.3) | 121 (64.7) | 3.09 | 1.95-4.88 | <0.001* | 3.32 | 1.99-5.53 | <0.001* |
| 50-59 | 163 (59.5) | 111 (40.5) | 2.47 | 1.62-3.77 | <0.001* | 2.61 | 1.67-4.08 | <0.001* |
| 60-69 | 195 (52.8) | 174 (47.2) | 1.89 | 1.26-2.82 | 0.002* | 1.93 | 1.26-2.95 | 0.011* |
| ≥70 | 51 (37.2) | 86 (62.8) | 1.00 | | | 1.00 | | |
| **Marital status** | | | | | | | | |
| Single | 43 (61.4) | 27 (38.6) | 2.07 | 1.15-3.72 | 0.015* | 1.78 | 0.95-3.31 | 0.070 |
| Married | 426 (56.1) | 333 (43.9) | 1.67 | 1.16-2.41 | 0.006* | 1.64 | 1.11-2.41 | 0.011* |
| Ever married | 61 (44.2) | 77 (55.8) | 1.00 | | | 1.00 | | |
| **Tribe** | | | | | | | | |
| Hill tribe | 18 (64.3) | 10 (35.7) | 1.50 | 0.68-3.28 | 0.310 | | | |
| Thai people | 512 (54.5) | 427 (45.5) | 1.00 | | | | | |
| **Thai ID card** | | | | | | | | |
| No | 7 (77.8) | 2 (22.2) | 2.91 | 0.60-14.08 | 0.184 | | | |
| Yes | 523 (54.6) | 435 (45.4) | 1.00 | | | | | |
| **Religion** | | | | | | | | |
| Buddhist | 528 (54.9) | 434 (45.1) | 1.82 | 0.30-10.97 | 0.511 | | | |
| Christian or Muslim | 2 (40.0) | 3 (60.0) | 1.00 | | | | | |
| **Education** | | | | | | | | |
| No education | 58 (47.5) | 64 (52.5) | 1.00 | | | | | |
| Primary school | 356 (54.9) | 293 (45.1) | 1.33 | 0.90-1.96 | 0.147 | | | |
| Secondary school and higher | 116 (59.2) | 80 (40.8) | 1.63 | 1.03-2.57 | 0.035* | | | |
| **Occupation** | | | | | | | | |
| Unemployed | 147 (51.9) | 136 (48.1) | 1.00 | | | | | |
| Agriculturist | 188 (53.4) | 164 (46.6) | 1.10 | 0.80-1.51 | 0.538 | | | |
| Employed | 130 (58.0) | 94 (42.0) | 1.31 | 0.92-1.87 | 0.127 | | | |
| Trade and government officer | 65 (60.2) | 43 (39.8) | 1.49 | 0.95-2.34 | 0.081 | | | |
| **Annual income** (baht) | | | | | | | | |
| ≤ 50,000 | 366 (55.1) | 298 (44.9) | 1.00 | | | | | |
| 50,001-100,000 | 97 (51.3) | 92 (48.7) | 0.88 | 0.64-1.22 | 0.470 | | | |
| ≥100,001 | 67 (58.8) | 47 (41.2) | 1.21 | 0.81-1.82 | 0.338 | | | |
| **Family debt** | | | | | | | | |
| No | 334 (59.0) | 232 (41.0) | 1.53 | 1.18-1.98 | 0.001* | | | |
| Yes | 196 (48.9) | 205 (51.1) | 1.00 | | | | | |
| **Family members** (people) | | | | | | | | |
| ≤ 4 | 444 (53.6) | 384 (46.4) | 1.00 | | | | | |
| ≥ 5 | 86 (61.9) | 53 (38.1) | 1.40 | 0.97-2.02 | 0.071 | | | |
| **Living with** | | | | | | | | |
| Alone | 28 (45.9) | 33 (54.1) | 0.69 | 0.36-1.30 | 0.257 | | | |
| Spouse | 330(56.0) | 259 (44.0) | 1.11 | 0.73-1.70 | 0.610 | | | |
| Child | 118 (54.6) | 98 (45.4) | 1.04 | 0.65-1.68 | 0.846 | | | |

*(Continued)*

**Table 4.** (Continued)

| Factors | Suboptimal glycemic control | | Univariable analysis | | | Multivariable analysis | | |
|---|---|---|---|---|---|---|---|---|
| | Yes (%) | No (%) | OR | 95% CI | p-value | AOR | 95% CI | p-value |
| Relatives | 54 (53.5) | 47 (46.5) | 1.00 | | | | | |

*Significance level α = 0.05

**Table 5. Identifying behavioral and psychological determinants associated with suboptimal glycemic control among DM patients by univariable and multivariable logistic regressions.**

| Factors | Suboptimal glycemic control | | Univariable analysis | | | Multivariable analysis | | |
|---|---|---|---|---|---|---|---|---|
| | Yes (%) | No (%) | OR | 95% CI | p-value | AOR | 95% CI | p-value |
| **Total** | **530 (54.8)** | **437 (45.2)** | **N/A** | **N/A** | **N/A** | **N/A** | **N/A** | **N/A** |
| **Smoking** | | | | | | | | |
| No | 423 (52.2) | 387 (47.8) | 1.00 | | | | | |
| Yes | 107 (68.2) | 50 (31.8) | 1.95 | 1.36-2.81 | <0.001* | | | |
| **Alcohol use** | | | | | | | | |
| No | 441 (54.2) | 372 (45.8) | 1.00 | | | | | |
| Yes | 89 (57.8) | 65 (42.2) | 1.11 | 0.79-1.58 | 0.526 | | | |
| **Exercise** | | | | | | | | |
| No | 301 (60.3) | 198 (39.7) | 1.00 | | | | | |
| Sometime | 75 (28.3) | 190 (71.7) | 3.85 | 2.79-5.31 | <0.001* | | | |
| Regular | 61 (30.0) | 142 (70.0) | 3.53 | 2.49-5.02 | <0.001* | | | |
| **Preparing food in daily life** | | | | | | | | |
| Themselves | 457 (53.5) | 398 (46.5) | 1.00 | | | | | |
| Buying | 73 (65.2) | 39 (34.8) | 1.56 | 1.03-2.34 | 0.033* | | | |
| **Type of rice eaten daily** | | | | | | | | |
| Non-sticky rice | 332 (50.4) | 327 (49.6) | 1.00 | | | 1.00 | | |
| Sticky rice | 198 (64.3) | 110 (35.7) | 1.77 | 1.34-2.34 | <0.001* | 1.61 | 1.19-2.61 | 0.002* |
| **Drinking tea** | | | | | | | | |
| No | 425 (51.8) | 395 (48.2) | 1.00 | | | | | |
| Yes | 105 (71.4) | 42 (28.6) | 2.32 | 1.58-3.40 | <0.001* | | | |
| **Drinking coffee** | | | | | | | | |
| No | 323 (48.9) | 337 (51.1) | 1.00 | | | | | |
| Yes | 207 (67.4) | 100 (32.6) | 2.16 | 1.62-2.86 | <0.001* | | | |
| **Stress** (ST-5) | | | | | | | | |
| Low | 390 (54.2) | 329 (45.8) | 1.00 | | | | | |
| Moderate | 90 (56.6) | 69 (43.4) | 1.07 | 0.75-1.51 | 0.691 | | | |
| High | 50 (56.2) | 39 (43.8) | 1.13 | 0.72-1.76 | 0.584 | | | |
| **Knowledge regarding DM prevention and control** | | | | | | | | |
| Low | 37 (47.4) | 41 (52.6) | 1.00 | | | | | |
| Moderate | 201 (52.3) | 183 (47.7) | 1.21 | 0.74-1.98 | 0.430 | | | |
| High | 292 (57.8) | 213 (42.2) | 1.51 | 0.94-2.45 | 0.087 | | | |
| **Attitudes toward DM prevention and control** | | | | | | | | |
| Poor | 122 (45.2) | 148 (54.8) | 1.00 | | | | | |
| Moderate | 329 (58.5) | 233 (41.5) | 1.70 | 1.26-2.27 | <0.001* | | | |
| Positive | 79 (58.5) | 56 (41.5) | 1.76 | 1.16-2.68 | 0.008* | | | |

*Significance level α = 0.05

**Table 6. Identifying medication and biochemical markers associated with suboptimal glycemic control among DM patients by univariable and multivariable logistic regressions.**

| Factors | Suboptimal glycemic control | | Univariable analysis | | | Multivariable analysis | | |
|---|---|---|---|---|---|---|---|---|
| | Yes (%) | No (%) | OR | 95% CI | p-value | AOR | 95% CI | p-value |
| Total | 530 (54.8) | 437 (45.2) | N/A | N/A | N/A | N/A | N/A | N/A |
| **Length of having DM** (year) | | | | | | | | |
| ≤ 10 | 362 (51.1) | 347 (48.9) | 1.00 | | | 1.00 | | |
| 11-20 | 113 (65.7) | 59 (34.3) | 1.72 | 1.22-2.43 | 0.002* | 1.98 | 1.37-2.86 | <0.001* |
| > 20 | 55 (64.0) | 31 (36.0) | 1.68 | 1.05-2.67 | 0.028* | 2.46 | 1.50-4.04 | <0.001* |
| **Experience of forgetting of taking a medicine** | | | | | | | | |
| No | 398 (50.8) | 385 (49.2) | 1.00 | | | 1.00 | | |
| Yes | 132 (71.7) | 52 (28.3) | 2.45 | 1.73-3.48 | <0.001* | 2.10 | 1.43-3.09 | <0.001* |
| **Having side effect of taking medicine** | | | | | | | | |
| No | 434 (51.9) | 402 (48.1) | 1.00 | | | | | |
| Yes | 96 (73.3) | 35 (26.7) | 2.43 | 1.61-3.65 | <0.001* | | | |
| **Medical expenses** | | | | | | | | |
| Covered by the national universal scheme | 519 (54.8) | 428 (45.2) | 0.99 | 0.40-2.41 | 0.986 | 2.67 | 1.00-7.08 | 0.049* |
| Self-paid | 11 (55.0) | 9 (45.0) | 1.00 | | | 1.00 | | |
| **History of having wound on foots** | | | | | | | | |
| No | 482 (53.1) | 425 (46.9) | 1.00 | | | | | |
| Yes | 48 (80.0) | 12 (20.0) | 3.17 | 1.69-5.94 | <0.001* | | | |
| **Diabetic nephropathy** | | | | | | | | |
| No | 191 (50.9) | 184 (49.1) | 1.00 | | | | | |
| Yes | 263 (54.1) | 223 (45.9) | 1.14 | 0.87-1.50 | 0.324 | | | |
| Do not know | 76 (71.7) | 30 (28.3) | 2.33 | 1.46-3.71 | <0.001* | | | |
| **Having HT** | | | | | | | | |
| No | 225 (57.0) | 170 (43.0) | 1.00 | | | | | |
| Yes | 263 (50.9) | 254 (49.1) | 0.76 | 0.59-1.00 | 0.050* | | | |
| Do not know | 42 (76.4) | 13 (23.6) | 2.41 | 1.25-4.64 | 0.008* | | | |
| **BMI** (kg/m$^2$) | | | | | | | | |
| Underweight (≤ 18.50) | 27 (50.9) | 26 (49.1) | 1.00 | | | | | |
| Normal weight (18.51-22.99) | 130 (45.9) | 153 (54.1) | 0.81 | 0.45-1.47 | 0.503 | | | |
| Overweight (≥ 23.00) | 373 (59.1) | 258 (40.9) | 1.39 | 0.79-2.44 | 0.248 | | | |
| **HDL cholesterol** (mg/dL) | | | | | | | | |
| Low (< 40) | 144 (55.0) | 118 (45.0) | 1.00 | 0.75-1.34 | 0.953 | | | |
| Normal (≥ 40) | 386 (54.8) | 319 (45.2) | 1.00 | | | | | |
| **LDL cholesterol** (mg/dL) | | | | | | | | |
| Normal (<100) | 272 (50.9) | 262 (49.1) | 1.00 | | | | | |
| High (≥100) | 258 (59.6) | 175 (40.4) | 1.46 | 1.13-1.89 | 0.003* | | | |
| **Triglycerides** (mg/dL) | | | | | | | | |
| Normal (<150) | 253 (52.1) | 233 (47.9) | 1.00 | | | | | |
| High (≥ 150) | 277 (57.6) | 204 (42.4) | 1.23 | 0.95-1.58 | 0.110 | | | |

*Significance level α = 0.05

## Discussion

Majority of the DM patients in Chiang Rai Province were female, aged 50-69 years, and had a poor economic and education status. Approximately, 6.6% smoked and used alcohol. A large proportion had poor knowledge and attitudes toward DM prevention and control. One-fourth

had been diagnosed with DM more than 10 years ago, one-fifth had experienced forgetting to take their medication, and almost all were covered by the universal medical scheme. Being older, married, forgetting to take medication, consuming sticky rice, having a long-term diagnosis of DM, and having a fully supported medical cost were found to be associated with suboptimal glycemic control among these DM patients.

In this study, the prevalence of suboptimal glycemic control among DM patients attending hospitals was very high (54.8%), which is similar to studies conducted in Iran (58.3%) [24], India (65.4%) [25], and Oman (54.0%) [26]. However, the prevalence of suboptimal glycemic control was found to be higher than that in a study conducted by Aekplakorn et al. [7] in Thailand, which was reported to be 30.0%. The differences might be impacted by the population's culture, especially the daily consumption of sticky rice [11] and living in a lower socioeconomic status [10] in northern Thailand compared to nationwide representation from a study conducted by Aekplakorn et al. [7]. The difference could also be impacted by the rescheduling of DM care and management during the COVID-19 pandemic [27, 28].

In our study, age was found to be associated with suboptimal glycemic control. Those who were younger were more likely to have suboptimal glycemic control than those who were older. This coincided with a study conducted in Pakistan, which showed that younger individuals were at a greater risk of having suboptimal glycemic control than older age groups [29]. However, a study conducted in Ghana [30] and Eastern Sudan [31] did not detect an association between age and suboptimal glycemic control. A study conducted in Saudi Arabia [32] reported that older age was significantly associated with suboptimal glycemic control. A study conducted in central Thailand reported that DM patients aged 60 years or older had a significantly greater chance of having suboptimal glycemic control than those aged less than 60 years [33]. The possible reason for suboptimal glycemic control among the people living in northern Thailand could be their busy farming life, and during the COVID-19 pandemic, the schedule for meeting with a doctor was extended to 3 or 6 months [34]. Moreover, many patients who have previously had good control of their blood glucose have been directed to receive their medication at health-promoting hospitals where no medical doctors or medical equipment for detecting blood glucose were available.

Married participants had a greater risk of suboptimal glycemic control than those who ever married. However, a study conducted in Ethiopia did not show a difference in suboptimal glycemic control between married and ever-married groups [35]. A study in Eastern Sudan reported that being unmarried had a greater risk of suboptimal glycemic control than being married [31]. A study in South Thailand reported that marital status was not associated with suboptimal glycemic control [36]. The people living in northern Thailand will have a busy daily life while being married due to working at their farm to support their family members. Therefore, married DM patients could have trouble following medical advice to control blood glucose.

Moreover, those who regularly consumed sticky rice were at a greater risk of having suboptimal glycemic control. This is the first report about an association between the type of rice consumed and suboptimal glycemic control. These results are supported by a study of the glycemic indexes of different types of rice in Thailand, which reported that sticky rice had the highest rapid available glucose (RAG) [37]. Sanmuangken et al. [38] reported that DM patients who consumed sticky rice were more likely to have suboptimal glycemic control than those who did not. Sticky rice consumption is common among people living in northern Thailand [11], and this is expected to be one of the most difficult issues to change in order to improve the blood glucose control among DM patients.

Haghighatpanah et al. [39] reported that those who had been diagnosed with DM more than 10 years ago had a greater chance of having poor glycemic control than those who had

been diagnosed with DM more recently. This was confirmed by studies in Ethiopia [40], Palestine [41], Malaysia [42], and Myanmar [43], which reported that DM patients who had been diagnosed more than 10 years ago had a greater risk of suboptimal glycemic control than those who had been diagnosed for less than 10 years. Moreover, a study in Thailand also reported that DM patients with 10 years or more after diagnosis had a greater chance of having suboptimal glycemic control than those who had been diagnosed less than 10 years prior [33].

In this study, it was found that those who did not adhere to their prescribed medication had a greater chance of having suboptimal glycemic control than those who did. A study in Ethiopia reported that DM patients with poor medication adherence were significantly associated with suboptimal glycemic control [44]. Moreover, Basu [45] noticed that medication adherence was a major contributing factor to suboptimal glycemic control. Gebrie et al. [35] reported that poor medication adherence was a significant contributing factor to suboptimal glycemic control. A study in Thailand reported that medication adherence was one of the significant factors for suboptimal glycemic control [46].

Those who were supported with all medical fees paid by the national scheme had a greater chance of having suboptimal glycemic control than those who were self-paying. This might be because those who did not have to pay medical fees were not concerned about taking their medication or strictly following the medical advice about their blood glucose management. Those who had to pay medical fees followed the medical advice. A systematic review [47] reported that the cost of diabetes treatment in low- and middle-income countries was $5 to $40 for a visit as an outpatient, and annual inpatient costs ranged from approximately $10 to over $1,000. The medical cost for DM treatment is very high for people in northern Thailand; however, most people have been supported under the universal coverage scheme.

Some limitations were detected throughout the study that might impact the analysis and interpretation of the findings. First, this study investigated some culture factors related to uncontrolled blood glucose, such as sticky rice consumption, which is a common dietary behavior among Thai people, but a few hill tribe people (2.9%) participated in the study, which might have little impact on the interpretation. Second, some questions ask about participants' past experiences, such as forgetting to take medications, having wounds on their feet, and experiencing side effects from taking medicine. Such questions might introduce recall bias, especially among those who were diagnosed a long time ago. Third, participants were not asked about the use of traditional herbs to heal their disease, which is practiced by some people and might interfere with HbA1c levels. Fourth, information on the prescription of regular DM drugs, which directly impact suboptimal glycemic control, was not collected in this study. Fifth, lifestyle modification should also be detected, especially those who were diagnosed several years previously, because lifestyle could also impact suboptimal glycemic control. Last, very few people in northern Thailand carry Hb variants [48], which might impact the interpretation of HbA1C. Thus, any further research should be aware of these points.

## Conclusions

A large proportion of DM patients in Chiang Rai people are facing a problem of uncontrolled blood glucose in their daily disease management. Several factors are associated with suboptimal glycemic control, including personal characteristics (age and marital status), sticky rice consumption, duration since DM diagnosis, forgetting to take medications and being given free access to care. Effective public health programs are needed to improve the quality of DM management systems among both patients and health care service systems. Among the elderly, having been diagnosed a long time ago and eating sticky rice regularly require the implementation of tools or innovations to improve the blood glucose control of these patients. Copayment

for medical fees should be carefully considered among DM patients. Some associated factors, such as eating sticky rice, need to be studied in more detail.

## Supporting information

**S1 Questionnaire.**
(DOCX)

**S1 Data.**
(XLSX)

## Acknowledgments

The author would like to thank all the participants for kindly providing all essential information regarding the research procedures.

## Author Contributions

**Conceptualization:** Fartima Yeemard, Peeradone Srichan, Tawatchai Apidechkul, Suphaphorn Utsaha.

**Data curation:** Fartima Yeemard, Peeradone Srichan, Tawatchai Apidechkul, Naphat Luerueang, Ratipark Tamornpark, Suphaphorn Utsaha.

**Formal analysis:** Fartima Yeemard, Peeradone Srichan, Tawatchai Apidechkul, Ratipark Tamornpark.

**Funding acquisition:** Tawatchai Apidechkul.

**Investigation:** Fartima Yeemard, Peeradone Srichan, Tawatchai Apidechkul, Naphat Luerueang, Ratipark Tamornpark, Suphaphorn Utsaha.

**Project administration:** Fartima Yeemard, Tawatchai Apidechkul.

**Supervision:** Tawatchai Apidechkul.

**Writing – original draft:** Fartima Yeemard, Peeradone Srichan, Tawatchai Apidechkul, Naphat Luerueang, Ratipark Tamornpark, Suphaphorn Utsaha.

**Writing – review & editing:** Fartima Yeemard, Peeradone Srichan, Tawatchai Apidechkul, Naphat Luerueang, Ratipark Tamornpark, Suphaphorn Utsaha.

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
