## [Decision Letter · Decision Letter 0]

21 Sep 2021

PONE-D-21-24649Prevalence and predictors of suboptimal glycemic control among patients with type 2 diabetes mellitus in northern Thailand: a hospital-based cross-sectional control studyPLOS ONE

Dear Dr. Apidechkul,

Thank you for submitting your manuscript to PLOS ONE. After careful consideration, we feel that it has merit but does not fully meet PLOS ONE’s publication criteria as it currently stands. Therefore, we invite you to submit a revised version of the manuscript that addresses the points raised during the review process.

The reviewers gave positive reviews; however, there are multiple issues especially in the Methods, Results and Discussion needed to be clarified.

We look forward to receiving your revised manuscript.

Kind regards,

Sompop Bencharit, DDS, MS, PhD, FACP

Academic Editor

PLOS ONE

Journal Requirements:

“The author is also grateful to all research assistants from the Center of Excellence for the Hill tribe Health Research for their help in data collection.”

“This research was supported by the Center of Excellence for the Hill tribe Health Research, Mae Fah Lung University, Thailand (Grant Number 1-2021). The funders had no role in study design, data collection and analysis, decision to publish, or preparation of the manuscript.”

Additional Editor Comments:

The reviewers gave positive reviews; however, there are multiple issues especially in the Methods, Results and Discussion needed to be clarified.

Reviewers' comments:

Reviewer's Responses to Questions

**Comments to the Author**

1. Is the manuscript technically sound, and do the data support the conclusions?

Reviewer #1: Partly

Reviewer #2: Yes

2. Has the statistical analysis been performed appropriately and rigorously? 

Reviewer #1: Yes

Reviewer #2: Yes

3. Have the authors made all data underlying the findings in their manuscript fully available?

Reviewer #1: Yes

Reviewer #2: Yes

4. Is the manuscript presented in an intelligible fashion and written in standard English?

Reviewer #1: Yes

Reviewer #2: Yes

5. Review Comments to the Author

Reviewer #1: 1. As sample size calculation; why do you use e as desired precision? Because the reference journal use d as absolute error or precision.

2. Please explain for P value in the sample size calculation, I could not find this figure in reference journal.

3. For validation questionnaire, why you do use 20 Thai people to validate this questionnaire? The Cronbanch’ s alpha showed 0.72 this is acceptable value, but it’s too low. What is the explanation for this low level?

4. For data gathering procedure, how do you believe in HbA1C level? Because HbA1C can’t use among hemoglobinopathy patients and hemoglobinopathy patients are common in the northern part of Thailand.

5. If you use HbA1C level to category the group of patients, how do you sure for one time of HbA1C test?

6. Ethics approval and consent to participate, how do you make a consent among minority group? Because the data also showed for non-Thai patients.

7. Do you exclude for additional medication or herbal that might interfere with HbA1C level? If not, this is important point as in questionnaire to identified cofounding factors to suboptimal glycemic?

8. As table 2, eating sticky rice is one factor to become suboptimal glycemic patients. Could you identify more than frequency of having sticky rice, such as amount of sticky rice.

9. As table 2, exercise is a factor to stimulate insulin releasing into the blood circulation. Could you add this information to identify as factor.

10. As table 4, How do you clarify for answer to experience of forgetting of taking medicine? Because this information is recall bias and not really true answer.

11. As table 4, I didn’t agree to use medical expense as one factor to be suboptimal glycemic, this should be a medicine type that patients received.

12. As table 4, diabetic nephropathy as do not know answer look to have statistically significant. What does it means?

Reviewer #2: This article is very interesting, and it is a valuable health service research article for diabetes outcome in Thailand. It is worth to consider publication of the article.

Although it is a cross-sectional study, it was well designed considering the power, and adjusting covariates.

Abstract

Conclusion highlighted the old age, but the multiple logistic regression results showed the higher number of adjusted odds ratios for age<49. Author may check carefully and revise the conclusion. Overall conclusion should be revised carefully in the abstract.

Background and literature review were well written.

Method is clearly written.

Important comments are

1. The level of HBA1C 7 is used as cut-off point. Authors need to explain with international guideline and Thai guideline.

2. When researchers target a treatment outcome, epidemiological study becomes prognostic in clinical research. Thinking as prognostic study authors must consider "treatment" that the patient is receiving at the time of recruitment for the study. 1. Types of the diabetes treatment such as oral hypoglycemic drugs, or insulin or life-style modification should be adjusted in the final model. 2. If the majority are on the oral hypoglycemic drugs, the type of agents and their effect are worth to consider as covariates. At least 1 should be performed to control the possible bias caused by medication. If those are not possible, authors should mention it in the limitation.

4. Adherence is of the highlight. Social support is important for adherence and diabetes lifestyle modification. I recommend reviewing and cite https://pubmed.ncbi.nlm.nih.gov/34299754/ it for discussion.

5. Author should mention the analysis especially how the final model was constructed. How each variable was decided to be included in the model. Table 4 is not informing how the variables are selected in term of concept, or in term of p-value to be in the multi-variable model.

6. ***Authors may divide the multi-variable models to be separate tables such as Table 5 andTable 6.

***Table 5 can have more than one models for the outcome of poor glycemic controls. Eg Model of social demographic, model of behavioral and psychological determinants, models of medication and biochemical markers

***Table 6 is to inform non-adherence and influencing factors.

These ***revisions are important to get published.

7. It is noticed that knowledge, attitude and stress-management were carefully measured. Author may present a model with carefully measured variables to report how those impact on the glycemic control or separate model for adherence. Revision may refer to above comment.

8. Reporting analysis results are qualified with 95% confidence interval values.

Results:

Characteristic of the sample and associations are well presented in tables and written adequately. Some comments in the analysis have already covered results.

Discussion:

1.The first sentence seems to reflect the sample characteristic. It is too dogmatic. I recommend writing simply just to describe the characters of the sample.

2.It is already well known that non-adherence will cause poor-glycemic control. What factors caused non-adherence in this study setting will be the interest for the reader. It would be more interesting to learn the factors associated to non-adherence of diabetes treatment. We recommend an additional model of logistic regression analysis and thorough discussion. Again, social support should be discussed.

Conclusion:

Authors made a strong conclusion to the consumption of sticky rice as a factor associated with poorer glycemic control. It should be noted that the exposure measurement is crude for sticky rice in this study, without the measurement of dose of exposure such as amount, frequency etc. We are hesitant to agree with a strong claim in the conclusion about sticky rice. It should be a recommendation for a further study finding out the association of sticky rice and diabetes in Thailand. Please revise the conclusion.

Otherwise, it is obvious that the authors had great effort in writing background and literature review and good discussion. English language is good standard. It is a carefully designed study with important impact on the public health. Just the final part of analysis needs reorientation. We should consider publication after a revision. I hope that my comments help to improve authors’ valuable work.

6. PLOS authors have the option to publish the peer review history of their article (what does this mean?). If published, this will include your full peer review and any attached files.

Reviewer #1: No

Reviewer #2: **Yes: **Myo Nyein Aung

---

## [Author Response · Author response to Decision Letter 0]

21 Oct 2021

Response to reviewers’ comments

Dear Editor and reviewers,

Thank you very much for such wonderful comments and suggestions which are greatly advantages to improve the manuscript. We have carefully revised and improved all comments accordingly. Many comments raised are very important. Some missing points in our study, We have put all these into the limitations.

Thank you so much for the great comments.

Regards,

TK

Journal Requirements:

: Thank you. We have carefully followed the instruction of PLOS ONE. 

“The author is also grateful to all research assistants from the Center of Excellence for the Hill tribe Health Research for their help in data collection.”

“This research was supported by the Center of Excellence for the Hill tribe Health Research, Mae Fah Lung University, Thailand (Grant Number 1-2021). The funders had no role in study design, data collection and analysis, decision to publish, or preparation of the manuscript.”

 : Thank you, we have deleted the information of funding and other information which might make confusing in acknowledgements.

Additional Editor Comments:

The reviewers gave positive reviews; however, there are multiple issues especially in the Methods, Results and Discussion needed to be clarified.

Reviewer #1: 1. As sample size calculation; why do you use e as desired precision? Because the reference journal use d as absolute error or precision.

: Sorry for making you confusing in using “e” or “d” which is different from the reference, we have replaced “d” in “e” as used in the reference. 

2. Please explain for P value in the sample size calculation, I could not find this figure in reference journal.

: The p-value in the reference mentioned is 0.05 in the reference is needed to be converted into the statistic value use in the formular which is Z =1.96. Then it would say that we are using the p-value in terms of tits statistics value (opposite value) in the sample size calculation.

While Z=1.96, then the p-value =0.05 in the standard normal distribution which is the same number but in opposite interpretation. Please see reference following

https://emj.bmj.com/content/emermed/17/6/409.full.pdf?__cf_chl_jschl_tk__=pmd_log0XXIITsRikSRqJsjyRRc99G6A8IQ3oLnpO6PzrEA-1632370178-0-gqNtZGzNAfujcnBszQ3R

3. For validation questionnaire, why you do use 20 Thai people to validate this questionnaire? The Cronbach’ s alpha showed 0.72 this is acceptable value, but it’s too low. What is the explanation for this low level?

: Thank you very much for the comment. Collecting data for 20 samples in the pilot phase (to validate the questionnaire) is one of the practical steps because if a set of questions in a questions either work or not, it will be detected within 20 samples. One more point with a number of questions in the questionnaire, 20 samples is enough to detect.

Basically, in the observational study especially in a population-based research, to set up the Cronbach’ s alpha between 0.7-0.8 is one most possible and advantage. To get over 0.8 is very difficult and also does not reflect the real situation. Setting up at 0.8 and over is a bit concrete, fix, and difficult to get the number. We had tried to increase this number to be 0.8 and over, but the number of the questions remaining in the research tools become very short, required a large sample in the pilot, and do not reflect well on the study samples characteristic. In addition, with more than 10 years research experience in doing observational study of us, set up at 0.7-0.8 (acceptable level in large sample size) is much more flexible to measure the variable related to human behaviors.

We always follow the information following reference which might help both of us understand my point; 

https://www.apjhs.com/index.php/apjhs/article/view/559/467

http://cda.psych.uiuc.edu/psychometrika_highly_cited_articles/cronbach_1951.pdf

: We do very hope that you understand us. 

4. For data gathering procedure, how do you believe in HbA1C level? Because HbA1C can’t use among hemoglobinopathy patients and hemoglobinopathy patients are common in the northern part of Thailand.

: Thank you for the comment. Using HbA1C is the standard guideline of clinical practice according to the WHO (https://www.who.int/diabetes/publications/report-hba1c_2011.pdf) and Ministry of Public Health Thailand to identify, monitor DM patients (https://www.dmthai.org/attachments/article/443/guideline-diabetes-care-2017.pdf)

:With these references, then we used HbA1C as the marker in the study. We do understand your point, but in clinical practice HbA1C is most used.

 : To make more for the point, we have put your concern on the limitation. 

: Moreover only 0.88% were found 14 Hb variants among people in northern Thailand, with a few people had a form that might interfere HbA1C (Panyasai S, Fucharoen G, Fucharoen S. Hemaglobin variants in northern Thailand: prevalence, heterogeneity and molecular characteristics. Genet Test Mol Biomarkers. 2016; 20(1): 37-43. 

5. If you use HbA1C level to category the group of patients, how do you sure for one time of HbA1C test?

: Basically, those who have been diagnosed for DM, they were monitored at least twice a year by HbA1C and by fasting blood glucose in every month before adjusting a prescription. Because HbA1C is used for assessing the remained glucose level which is a bit better than fasting blood glucose. Sometime, the patients do not strictly follow the prescription, but able to seriously restrict their eating behavior a few days before meeting a doctor to make sure that fasting blood glucose is met the satisfaction of a doctor. Then, the HbA1C is better to use for monitoring.

In addition, in our inclusion criterion page 4, lines 7-9, is clearly defined the inclusion criteria that at least two years of the diagnosis required to participate the study.

6. Ethics approval and consent to participate, how do you make a consent among minority group? Because the data also showed for non-Thai patients.

: Almost of the non-Thai people requited into the study, they could use Thai fluently. Only three cases who could not use Thai fluent, the information regarding the study were completed by the help of their relative who spoke Thai. Even being Thai people, some people could not read Thai, then they were asked in fingerprinting to the consent from after having been clearly explained. Please see the statement on the topic of ethic in page 6, lines 17-22. 

7. Do you exclude for additional medication or herbal that might interfere with HbA1C level? If not, this is important point as in questionnaire to identified cofounding factors to suboptimal glycemic?

: Thank you for such great comment. We did not ask questions about using any herbs along the treatment to individual. We have added into the limitations, page 21, lines 19-21. 

8. As table 2, eating sticky rice is one factor to become suboptimal glycemic patients. Could you identify more than frequency of having sticky rice, such as amount of sticky rice.

: So sorry, with the preliminary research objectives in identifying all possible risk factors associated with suboptimal glycemic, and the nature of a cross-sectional, we did not focus on the detail of any specific factor or behavior such as sticky rice eating. However, thank you very much for the comment, we have planned to work in the point oi our next project. 

9. As table 2, exercise is a factor to stimulate insulin releasing into the blood circulation. Could you add this information to identify as factor.

:Thank you the comment, we have added information of exercise in table 

10. As table 4, How do you clarify for answer to experience of forgetting of taking medicine? Because this information is recall bias and not really true answer.

: Thank you for the great notice. Asking questions on experience of forgetting of taking medicine, we used two question 15 and 16, as following;

“Q15. Have you forget taking diabetes medication last week? �Yes �No

Q16. Have you forget taking diabetes medication last month? �Yes �No”

: We agree with you that it could lead to have recall bias, then we added in a limitation, please see page 22, lines 17-19.

11. As table 4, I didn’t agree to use medical expense as one factor to be suboptimal glycemic, this should be a medicine type that patients received.

: Thank you for the great comment. However, as in Thailand, there is not everyone who has been grated to access medical fee without charging. Then, at the step of literature review, we found that an affordable was one of the factors contributing to the suboptimal glycemic. Then, it had been induced in the set of questionnaire and in the model.

: However, we totally agree with you on the type of medicines, unfortunately the information was not collected. Then we put this point into the limitation of the study. Please see page 22, lines 

12. As table 4, diabetic nephropathy as do not know answer look to have statistically significant. What does it means?

: In terms of “Do not know” means that the participants (DM patients) did not know their status of diabetic nephropathy. Even the statistic significance found in the univariate analysis, but it’s not in the multivariable model.

Reviewer #2: This article is very interesting, and it is a valuable health service research article for diabetes outcome in Thailand. It is worth to consider publication of the article. Although it is a cross-sectional study, it was well designed considering the power, and adjusting covariates.

Abstract

Conclusion highlighted the old age, but the multiple logistic regression results showed the higher number of adjusted odds ratios for age<49. Author may check carefully and revise the conclusion. Overall conclusion should be revised carefully in the abstract.

: Thank you for the great notice. We have revised the conclusion, see page 2, lines 13-14.

Background and literature review were well written.

: Thank you.

Method is clearly written.

: Thank you.

Important comments are

1. The level of HBA1C 7 is used as cut-off point. Authors need to explain with international guideline and Thai guideline.

: Thank you for the concern, we had looked carefully during proposal development on the cutoff the HbA1C which is found the that standard of the American Diabetes Association, reference no.24, is used widely including the WHO has recommended. Then we used this standard as the cut off. 

2. When researchers target a treatment outcome, epidemiological study becomes prognostic in clinical research. Thinking as prognostic study authors must consider "treatment" that the patient is receiving at the time of recruitment for the study. 

: Thank you for the comment. We totally agreed with your point and it is one of the mistakes of our study that did not get information of the prescriptions. We have added this in the limitation. Thank you so much!

2.1 Types of the diabetes treatment such as oral hypoglycemic drugs, or insulin or life-style modification should be adjusted in the final model. 

: Thank you for the great comment. However, as response previously, we did not collect information about drugs prescribes individually including insulin and we have added these points in to the limitation. 

: Since a cross-sectional was used in this study, which means that a snap shot data collection was executed, then collecting information of the behavioral modification was not available. We did not do prospective study which could be able to collect data on the life-style modification. 

: Thank you so much, we have kept it into the limitation for further study. 

2.2 If the majority are on the oral hypoglycemic drugs, the type of agents and their effect are worth to consider as covariates. At least 1 should be performed to control the possible bias caused by medication. If those are not possible, authors should mention it in the limitation.

: Thank you for the comment. We have added the point in limitation, please see page 22, lines 19-20.

3. Adherence is of the highlight. Social support is important for adherence and diabetes lifestyle modification. I recommend reviewing and cite https://pubmed.ncbi.nlm.nih.gov/34299754/ it for discussion.

: Thank you for the suggestion. We aimed to detect the factors associated with of suboptimal glycemic control by a cross sectional which was not focus on a particularly social support for adherence and diabetes life style modification. However, these point are great to study especially those people living northern Thailand. To make readers having most benefits, we have added into the recommendation for further study. Please see in page 22, lines 20-22.

4. Author should mention the analysis especially how the final model was constructed. How each variable was decided to be included in the model. Table 4 is not informing how the variables are selected in term of concept, or in term of p-value to be in the multi-variable model.

: Thank you, we have added information of executing the analysis in page 6, lines 14-20.

5. ***Authors may divide the multi-variable models to be separate tables such as Table 5 and Table 6.

***Table 5 can have more than one models for the outcome of poor glycemic controls. Eg Model of social demographic, model of behavioral and psychological determinants, models of medication and biochemical markers

***Table 6 is to inform non-adherence and influencing factors.

These ***revisions are important to get published.

: Thank you for the great comment, we have divided into three univariable and multivariable analysis for social demographic to the Table 4, model of behavioral and psychological determinants to the Table 5, and models of medication and biochemical markers to the Table 6.

6. It is noticed that knowledge, attitude, and stress-management were carefully measured. Author may present a model with carefully measured variables to report how those impact on the glycemic control or separate model for adherence. Revision may refer to above comment.

: Thank you again for the comment. As response you in previous comment that after discussion with team and statisticians to have all variables into the same model to predict the outcome would be better in term of the power of the statistics (1-�) by reducing �-error in each step of the model including to compliance the nature of a cross-sectional study using in the study.

7. Reporting analysis results are qualified with 95% confidence interval values.

: Thank you, it is because we collect large samples and the interval a bit narrow which is presented a high precision of the estimation. 

Results:

Characteristic of the sample and associations are well presented in tables and written adequately. Some comments in the analysis have already covered results.

: Thank you.

Discussion:

1.The first sentence seems to reflect the sample characteristic. It is too dogmatic. I recommend writing simply just to describe the characters of the sample.

: Thank you for the comment, we have revised it. 

2.It is already well known that non-adherence will cause poor-glycemic control. What factors caused non-adherence in this study setting will be the interest for the reader. It would be more interesting to learn the factors associated to non-adherence of diabetes treatment. We recommend an additional model of logistic regression analysis and thorough discussion. Again, social support should be discussed.

: Thank you so much for the comment. Again, we aimed to investigate the factors associated with suboptimal glycemic control among the DM patients in Chiang Rai. We did not focus our objective to factors adherence to the suboptimal glycemic control, however, it is excellent point for further study. We will keep go on to make more understand the factors related to adherence of access DM clinic and follow treatment guideline of DM patients. 

Conclusion:

Authors made a strong conclusion to the consumption of sticky rice as a factor associated with poorer glycemic control. It should be noted that the exposure measurement is crude for sticky rice in this study, without the measurement of dose of exposure such as amount, frequency etc. We are hesitant to agree with a strong claim in the conclusion about sticky rice. It should be a recommendation for a further study finding out the association of sticky rice and diabetes in Thailand. Please revise the conclusion.

: Thank you for the great comment. We have revised the point of conclusion, page 22 (lines 31-33) – 23 (lines 1-2)

Otherwise, it is obvious that the authors had great effort in writing background and literature review and good discussion. English language is good standard. It is a carefully designed study with important impact on the public health. Just the final part of analysis needs reorientation. We should consider publication after a revision. I hope that my comments help to improve authors’ valuable work.

: Thank you

TK

Assist Prof. Dr. Tawatchai Apidechkul, MSc (Infectious Epidemiology), Dr.P.H (Epidemiology)

Dean, School of Health Science, MFU

Director, Center of Excellence of Hill Tribe Health Research, WHO-CC 

Former Hubert H Humphrey Fellow (2013-2014), Emory University

Global Health Delivery Intensive (Harvard T.H. Chan School of Public Health)

---

## [Editor Report · Decision Letter 1]

3 Jan 2022

Prevalence and predictors of suboptimal glycemic control among patients with type 2 diabetes mellitus in northern Thailand: a hospital-based cross-sectional control study

PONE-D-21-24649R1

Dear Dr. Apidechkul,

We’re pleased to inform you that your manuscript has been judged scientifically suitable for publication and will be formally accepted for publication once it meets all outstanding technical requirements.

Kind regards,

Sompop Bencharit, DDS, MS, PhD, FACP

Academic Editor

PLOS ONE

Additional Editor Comments (optional):

Thank you for submitting the revision and response to reviewer's comments.
---

## [Editor Report · Acceptance letter]

6 Jan 2022

PONE-D-21-24649R1 

Prevalence and predictors of suboptimal glycemic control among patients with type 2 diabetes mellitus in northern Thailand: a hospital-based cross-sectional control study 

Dear Dr. Apidechkul:

I'm pleased to inform you that your manuscript has been deemed suitable for publication in PLOS ONE. Congratulations! Your manuscript is now with our production department. 

Kind regards, 

on behalf of

Dr. Sompop Bencharit 

Academic Editor

PLOS ONE